# AdaEmbed: Semi-supervised Domain Adaptation in the Embedding Space

## Abstract

Semi-supervised domain adaptation (SSDA) presents a critical hurdle in computer vision, especially given the frequent scarcity of labeled data in real-world settings. This scarcity often causes foundation models, trained on extensive datasets, to underperform when applied to new domains. AdaEmbed, our newly proposed methodology for SSDA, offers a promising solution to these challenges. Leveraging the potential of unlabeled data, AdaEmbed facilitates the transfer of knowledge from a labeled source domain to an unlabeled target domain by learning a shared embedding space. By generating accurate and uniform pseudo-labels based on the established embedding space, the model overcomes the limitations of conventional SSDA, thus enhancing performance significantly. Our method's effectiveness is validated through extensive experiments on benchmark datasets such as DomainNet, Office-Home, and VisDA-C, where AdaEmbed consistently outperforms all the baselines, setting a new state of the art for SSDA. With its straightforward implementation and high data efficiency, AdaEmbed stands out as a robust and pragmatic solution for real-world scenarios, where labeled data is scarce. To foster further research and application in this area, we are sharing the codebase of our unified framework for semi-supervised domain adaptation [URL will be shared upon publication].

## 1 Introduction

Foundation models, renowned for their extensive pre-training on vast datasets, have established new standards in a variety of computer vision tasks. However, despite their advanced architecture and training, these models often encounter challenges in real-world scenarios characterized by a significant divergence between the target domain and the source domain of their initial training. This phenomenon, known as domain shift, frequently results in a noticeable drop in the performance of these models, underscoring the need for robust domain adaptation strategies Donahue et al. (2014); Tzeng et al. (2017). The domain adaptation aims to transfer knowledge from a labeled source domain to a new but related target domain. Acquiring labeled data for the target domain can be challenging and resource-intensive, underscoring the need for semi-supervised domain adaptation (SSDA). SSDA leverages a small amount of labeled data in the target domain and a larger amount of unlabeled data to enhance the model's performance. Despite its wide range of real-world applications, SSDA has received less attention than both unsupervised domain adaptation (UDA) and semi-supervised learning (SSL).

Existing Semi-supervised Domain Adaptation (SSDA) research primarily focuses on two strategies: learning domain-invariant embeddings and generating pseudo-labels for unlabeled target examples. Notable works include Saito et al. (2018), which emphasizes discrepancy maximization for alignment, and others like Ganin et al. (2016); Sun & Saenko (2016), which propose matching feature distributions in deep networks. Saito et al. (2019) introduced an adaptation technique based on conditional entropy, while Berthelot et al. (2021) developed AdaMatch for confident pseudo-label generation. Most current SSDA methods enhance pseudo-labels through prediction refinements or utilize contrastive learning for alignment at both inter-domain and instance levels.

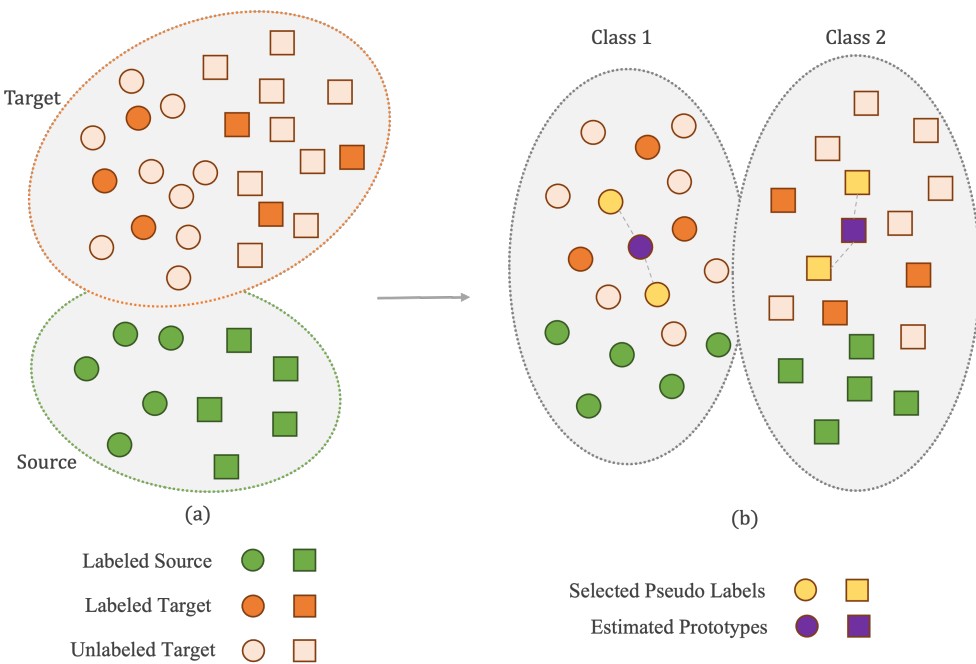

Figure 1: AdaEmbed method for semi-supervised domain adaptation (SSDA). The embedding space for a general SSDA problem is shown in (a), while (b) illustrates how AdaEmbed estimates prototypes based on labeled and unlabeled samples in the feature space, selects unlabeled samples in the embedding space to generate pseudo-labels given their proximity to prototypes, and trains the model on both labels and pseudo-labels. In addition, a contrastive feature loss is incorporated during training to learn a more effective shared embedding for both source and target domains.

In our paper, we present AdaEmbed, a new SSDA methodology that emphasizes learning a shared embedding space and prototypes for generating precise and balanced pseudo-labels for the unlabeled target domain (Figure 1). AdaEmbed incorporates two primary forms of training supervision: cross-entropy loss for actual and pseudo-labels, and contrastive loss for aligning instance features. This approach fosters a robust embedding space that is resilient to distribution shifts. Diverging from previous models, AdaEmbed uniquely leverages the embedding space and prototypes to produce more evenly distributed pseudo-labels, effectively addressing the imbalance issue prevalent in earlier SSDA methods Singh (2021); Sahoo et al. (2021).

We conduct thorough experiments on major domain adaptation benchmarks, including DomainNet-126, Office-Home, and VisDA-C, to evaluate the efficacy of AdaEmbed for SSDA. Our primary contributions in this work are:

- We have proposed AdaEmbed as a comprehensive approach for domain adaptation. This method is versatile, applicable to both UDA and SSDA scenarios. It effectively leverages a small number of labeled images in the target domain, in addition to the larger set of unlabeled data.

- A key feature of AdaEmbed is its innovative pseudo-labeling strategy, which uses the embedding features to generate accurate and balanced pseudo-labels. This approach addresses the common issue of imbalanced pseudo-label distributions in SSDA.

- Our experiments demonstrate AdaEmbed's superiority over all baseline models. A detailed ablation study further validates its efficacy, highlighting the critical roles of pseudo-labeling, contrastive loss, and entropy loss in enhancing the method's effectiveness.

- AdaEmbed is designed to be model-agnostic, allowing integration with any standard classification backbone. This flexibility extends its applicability, making it a versatile tool in various computer vision applications.

## 2 Related Works

### 2.1 Unsupervised Domain Adaptation (UDA)

Unsupervised Domain Adaptation (UDA) involves generalizing a model trained on a source domain to a new but related, fully unlabeled target domain. A common focus in UDA literature Liang et al. (2019); Long et al. (2015; 2016) is reducing the discrepancy between source and target domain representations. Some approaches align the final representations using maximum mean discrepancy Tzeng et al. (2014); Gretton et al.; Xiao & Zhang (2021), while others match the distribution of intermediate features in deep networks Sun & Saenko (2016); Peng et al. (2019); Chen et al. (2018). In Maximum Classifier Discrepancy (MCD) Saito et al. (2018), two task classifiers are trained to maximize the discrepancy on the target sample, then the feature generator is trained to minimize this discrepancy.

Adversarial learning-based approaches Ganin & Lempitsky (2015); Kang et al. (2018); Tzeng et al. (2017); Zhang et al. (2018a;b) use domain discriminators and generative networks to introduce ambiguity between source and target samples, learning domain-invariant features in the process. Several methods also use pseudo-labels Bruzzone & Marconcini (2010); Chen et al. (2011); Xie et al. (2018); Long et al. (2013); Saito et al. (2017) to incorporate categorical information in the target domain. Test-time adaptation approaches are another branch of research that utilizes the source model and unlabeled target domain data, with notable methods including AdaContrast Chen et al. (2022), which specifically refines pseudo-labels through soft voting among nearest neighbors in the target feature space, SHOT Liang et al. (2020), and TENT Wang et al. (2021).

### 2.2 Semi-supervised Domain Adaptation (SSDA)

Semi-supervised Domain Adaptation (SSDA) differs from UDA in that it utilizes a labeled set in the target domain. The goal remains to achieve the highest performance on the target domain using both labeled and unlabeled data from the source and target domains. SSDA is less explored than UDA, with most research focusing on image classification tasks Yao et al. (2015); Ao et al. (2017); Saito et al. (2019). A key methodology in this area is the Minimax Entropy (MME) approach Saito et al. (2019) which focuses on adaptation by alternately maximizing the conditional entropy of unlabeled target data concerning the classifier and minimizing it with respect to the feature encoder. Another notable contribution is AdaMatch, proposed by Berthelot et al. (2021), which extends the principles of FixMatch Sohn et al. (2020) to SSDA setting. Additionally, recent advancements in SSDA include the introduction of CLDA Singh (2021), a method utilizing contrastive learning for reducing discrepancies both within and between domains, and ECACL Li et al. (2021), which employs robust data augmentation to refine alignment strategies in SSDA scenarios. Building on the core principles established by previous research, our AdaEmbed framework capitalizes on the embedding space to propose a comprehensive approach to domain adaptation.

### 2.3 Semi-Supervised Learning (SSL)

Semi-Supervised Learning (SSL) leverages both unlabeled and partially labeled data to train models. Recent advances in self-supervised learning Caron et al. (2020); Chen et al. (2020a;b); He et al. (2020); Grill et al. (2020) enable model pre-training on large-scale datasets, followed by fine-tuning using a small labeled dataset. Consistency regularization-based approaches Sajjadi et al. (2016); Laine & Aila (2016); Tarvainen & Valpola (2017) are widely used in SSL to enforce the consistency between different augmentations of the same training sample. Augmentation-based methods like MixMatch Berthelot et al. (2019b), ReMixMatch Berthelot et al. (2019a), and FixMatch Sohn et al. (2020) utilize pseudo-labeling and consistency regularization. However, these methods cannot be directly applied to the SSDA setting due to their assumption that data is sampled from the same distribution.

### 2.4 Few-Shot Learning (FSL)

Few-Shot Learning (FSL) tasks require the model to generalize from limited training samples. A rich line of work exists on meta-learning for few-shot learning, with recent works focusing on optimization-based, metric-

based, and model-based approaches. Optimization-based methods Finn et al. (2017); Jamal & Qi (2019); Ravi & Larochelle (2016); Li et al. (2017); Rusu et al. (2019); Sun et al. (2019) aim to find the best model parameters for rapid adaptation to unseen tasks. Metric-based methods Vinyals et al. (2016); Sung et al. (2017); Snell et al. (2017); Oreshkin et al. (2018); Koch et al. (2015); Wang et al.; Satorras & Estrach (2018) learn an embedding function using a distance function within the meta-learning paradigm. Transductive learning-based approaches Dhillon et al. (2020); Bateni et al. (2020); Liu et al. (2019); SHEN et al. have also been proposed for FSL. Finally, model-based approaches Santoro et al. (2016); Cai et al. (2018); Munkhdalai et al. (2017); Mishra et al. (2018); Munkhdalai & Yu (2017); Edwards & Storkey (2017) involve designing architectures specifically tailored for fast adaptation, for instance, through the use of external memory. Our proposed approach, AdaEmbed, draws inspiration from the metric-based methods.

## 3 Method

We propose AdaEmbed, a new domain adaptation approach for both unsupervised domain adaptation (UDA) and semi-supervised domain adaptation (SSDA) settings on any classification task. Our new method utilizes the embedding space to generate pseudo-labels for the unlabeled target samples and with the help of contrastive learning learns an embedding shared between source and target domains. The main difference between UDA and SSDA is that in SSDA setting we have a few labeled images (samples) in the target domain. Therefore in the source domain we are given source images and their corresponding labels $\mathcal{D}_s = \{(x_s^i, y_s^i)\}_{i=1}^{n_s}$. In the target domain, we are also given a limited set of labeled target samples $\mathcal{D}_t = \{(x_t^i, y_t^i)\}_{i=1}^{n_t}$. The rest of the target is unlabeled and is denoted by $\mathcal{D}_u = \{x_u^i\}_{i=1}^{n_u}$. Our goal is to train the model on $\mathcal{D}_s$, $\mathcal{D}_t$, and $\mathcal{D}_u$ and evaluate on the test set of the target domain. In UDA settings we only have access to $\mathcal{D}_s$ and $\mathcal{D}_u$, therefore $\mathcal{D}_t$ will be empty. Our approach is not limited to only images and can take any input type.

### 3.1 AdaEmbed

In this subsection we discuss different component of our method AdaEmbed. Data points are first augmented and passed through the encoder and momentum encoders. Then we generate pseudo labels for unlabeled target samples based on the embedding features. The features are contrasted in the embedding space to push the similar labels/pseudo-labels together and different class apart from each other. Finally, the entropy of unlabeled target predictions is used to update the prototypes in our model.

**Model.** Our base model architecture, following Chen et al. (2019), consists of a general encoder (feature extractor) $F(.) : \mathcal{X}_s \to \mathbb{R}^d$ and a single layer classifier $C(.) : \mathbb{R}^d \to \mathbb{R}^c$ where $d$ and $c$ are feature dimension and number of classes. Any deep learning based neural architecture can be used as the feature extractor for image or video inputs ($x$). The extracted features $f = F(x) \in \mathbb{R}^d$ are first normalized with $l_2$ normalization i.e. $\frac{F(x)}{||F(x)||}$. The classifier consist of a weight matrix $W = [w_1, w_2, \ldots, w_c] \in \mathbb{R}^{d \times c}$ and it outputs logits

$$z = C(x) = \frac{W^T F(x)}{||F(x)||}$$

The logits are then passed through a softmax to generate the probabilistic output $p = softmax(z)$. The classifier uses cosine distances between the input feature and the weight vector for each class that aims to reduce intra-class variations. As discussed in Chen et al. (2019), the weight vectors can be described as estimated prototypes for each class and the classification is based on the distance of the input feature to these learned prototypes based on cosine similarity.

Consider each sample and its augmented version as $x$ and $\tilde{x}$ respectively. For images, we use RandAugment Cubuk et al. (2020) and for videos we augment them frame-wise with the same augmentation strategy. The augmented data point $\tilde{x}$ is passed through the encoder to extract the features $\tilde{f} = F(\tilde{x})$. We also keep the original version and pass it though the momentum encoder to keep a queue of extracted features that will be later used for generating pseudo-labels and computing the contrastive losses. The momentum encoder $F'$ parameters $\theta_{F'}$ are initialized with the encoder weights $\theta_F$ at the beginning of training, and updated at

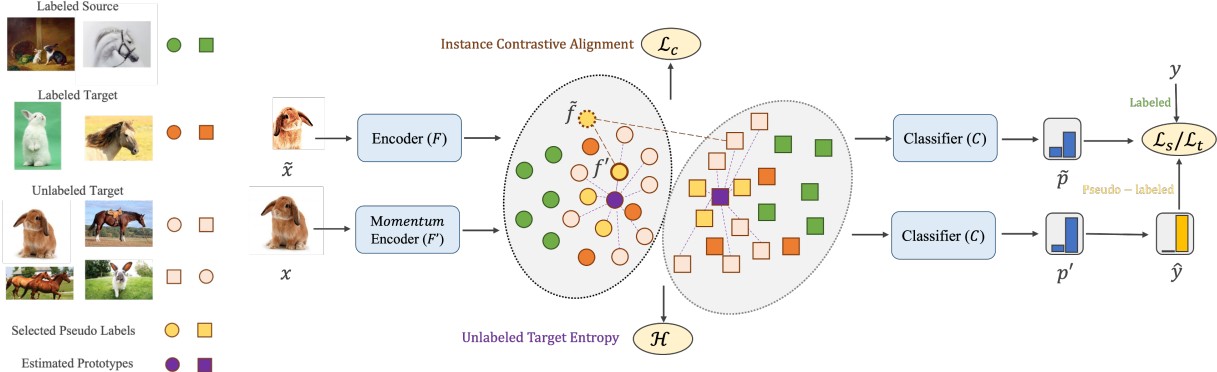

Figure 2: The framework for our semi-supervised domain adaptation method (AdaEmbed). The left column visualises the types of data points and their corresponding symbol in this figure. At the beginning of adaptation process both the encoder and momentum encoder are initialized with the model trained on labeled data. At each iteration we augment the data and send the augmented version ($\tilde{x}$) to our encoder while the original version ($x$) is passed through the momentum encoder. The features generated by the encoders are then given to the classifier to generate the predictions ($p$ and $\tilde{p}$). For labeled samples we compute the cross entropy loss between the given label and the corresponding prediction ($\mathcal{L}_s$). For unlabeled target samples we first generate pseudo-label ($\hat{y}$) for a few samples based on their proximity to the class prototypes, and use the pseudo-labels for the target loss ($\mathcal{L}_t$). We also unitize an instance contrastive alignment loss ($\mathcal{L}_c$) to push the original and augmented features ($f'$ and $\tilde{f}$) together while pushing features with different labels/pseudo-labels apart. Finally the prototypes are updated based on the entropy of unlabeled target predictions ($\mathcal{H}$). The training algorithm for AdaEmbed is outlined in Algorithm 1.

each minibatch following He et al. (2020):

$$\theta_{F'} \leftarrow m\theta_{F'} + (1-m)\theta_F \tag{1}$$

Then the momentum features and probabilities are computed as:

$$f' = F'(x), \quad p' = softmax(C(f'))$$

The momentum features and probabilities are stored in a queue $Q = \{(f'^i, p'^i)\}_{i=1}^M$ of size $M$. This memory is used to generated pseudo-labels for unlabeled samples and to computer contrastive loss for features. Please note that we only use the features and probabilities corresponding to the augmented data ($\tilde{f}, \tilde{p}$) in back-propagating gradients.

**Supervised Loss.** For the labeled data, we compute the standard cross entropy loss which forms the supervised term in our final loss function:

$$\mathcal{L}_s = \mathbb{E}_{(x,y)\in\mathcal{D}_s\cup\mathcal{D}_t}\mathcal{L}_{ce}(y, \tilde{p}) \tag{2}$$

**Pseudo-Labeling.** In AdaEmbed, we propose a new pseudo-labeling strategy for generating accurate and balanced pseudo-labels based on the embeddings. The proximity to the class prototype (weights of the matrix $W$ in classifier $C$) is used as a measure of accuracy of pseudo-labels. We also sample the same number of pseudo-labels for each prototype to ensure a nearly uniform distribution of generated pseudo-labels. After adding the new sample (or mini-batch of samples) ($f', p'$) to the memory $Q$ we select the $k$ nearest neighbours of each prototype, where $k$ is a hyper-parameter and lower $k$ results in more accurate pseudo-labels. If the sample feature $f'$ falls inside one of the k nearest neighbours, we generate pseudo-labels for it otherwise it will be ignored. More formally,

$$mask = (f' \in knn(Q_f)) \tag{3}$$

where $Q_f$ are the features in the memory queue. If selected, the generated pseudo-labels will be $\hat{y} = \arg\max(p')$ Selecting the $k$ nearest neighbours for each class prototype ensures the most accurate pseudo-labels are selected and the generated pseudo-labels overall have a uniform distribution which is critical in datasets with long-tailed distribution.

Based on generated pseudo-labels, we add a new target loss term as follows:

$$\mathcal{L}_t = \mathbb{E}_{x \in \mathcal{D}_u}[\mathcal{L}_{ce}(\text{stopgrad}(\hat{y}), \tilde{p}) \cdot mask] \tag{4}$$

Here, stopgrad stops the gradients from back-propagating into the momentum encoder branch.

**Instance Contrastive Alignment.** For unlabeled target samples, we use the InfoNCE loss to contrast the generated features with the features in the memory bank Oord et al. (2018):

$$\mathcal{L}_c = -\mathbb{E}_{x \in \mathcal{D}_u} \log \frac{\exp(sim(\tilde{f} \cdot f')/\tau)}{\sum_{q \in \bar{Q}_f(\hat{y})} \exp(sim(\tilde{f} \cdot q)/\tau)} \tag{5}$$

However in contrast to Oord et al. (2018), we only use the features in the feature bank where their corresponding pseudo-labels are not similar, i.e:

$$\bar{Q}_f(\hat{y}) = \{q|(q, p) \in Q, \arg\max(p) \neq \hat{y}\} \tag{6}$$

In this way we are only pushing away target samples with different pseudo-labels in the embedding space, therefore preserving the clusters for each class in the embedding space while maximizing the distances between them.

**Updating Prototypes.** The prototypes are learned based on adversarial minimax entropy training using unlabeled target examples. The position of prototypes are updated by maximizing the entropy between wight matrix $W$ and unlabeled target features $\tilde{f}$:

$$\mathcal{H} = -\mathbb{E}_{x \in \mathcal{D}_u} \sum_{i=1}^{c} \log \tilde{p}_i \log(\tilde{p}_i) \tag{7}$$

where $\tilde{p}_i$ is the $i$th dimension of $\tilde{p}$, i.e. the probability of prediction to class $i$. The higher entropy value means each $w_i$ (prototypes) is similar to target features and therefore well positioned to be domain-invariant features for each class.

**Training Objectives.** The full overview of our domain adaptation method is depicted in Fig. 1. As described in this section, in AdaEmbed we have three forms of training objectives: 1) cross-entropy loss for the labels of the labeled samples 2) masked cross-entropy loss for generated pseudo-labels of the unlabeled target 4) contrastive loss for the features of the unlabeled data.

To sum up the discussion, the model ($F$ and $C$ combined) is optimized to minimize the supervised loss $\mathcal{L}_s$ for labeled data points, and minimize the target pseudo-labeling loss $\mathcal{L}_t$ and contrastive loss $\mathcal{L}_c$ for unlabeled samples. However, the task classifier $C$ is trained to maximize the entropy $\mathcal{H}$ while the feature extractor $F$ is trained to minimize it. The overall adversarial learning objectives can be formulated as:

$$\hat{\theta}_F = \arg\min_{\theta_F} \mathcal{L}_s + \lambda_t \mathcal{L}_t + \lambda_c \mathcal{L}_c + \lambda \mathcal{H} \tag{8}$$

$$\hat{\theta}_C = \arg\min_{\theta_C} \mathcal{L}_s + \lambda_t \mathcal{L}_t + \lambda_c \mathcal{L}_c - \lambda \mathcal{H} \tag{9}$$

As suggested by Saito et al. (2019), we use a gradient reversal layer Ganin & Lempitsky (2015) to to flip the gradient between $F$ and $C$ with respect to $\mathcal{H}$ and convert the minimax training with one forward and back-propagation. We are ready to present Algorithm 1 for Semi-Supervised Domain Adaptation.

---

**Algorithm 1** Semi-Supervised Domain Adaptation in the Embedding Space

---

**Input:** Source dataset $\mathcal{D}_s$, labeled target dataset $\mathcal{D}_t$, Unlabeled target dataset $\mathcal{D}_u$.

**Initialize:** Momentum encoder $F'$ with encoder $F$ pretrained on labeled samples. Memory $Q = \{\}$

 1: **repeat**
 2:     Sample labeled source sample $(x_s, y_s) \in \mathcal{D}_s$, labeled target sample $(x_t, y_t) \in \mathcal{D}_t$, and unlabeled target input $x_u \in \mathcal{D}_u$.
 3:     Apply augmentation on inputs to get $\tilde{x}_s$, $\tilde{x}_t$, and $\tilde{x}_u$
 4:     Compute supervised cross entropy loss for augmented labeled data using Eq. 2
 5:     Pass the original data points through the momentum encoder $F'$ to get features $f'_s$, $f'_t$, and $f'_u$ and then classifier $C$ to get predictions $p'_s$, $p'_t$, and $p'_u$.
 6:     Generate pseudo-label $\hat{y}_u = \arg\max(p'_u)$ if $f'_u$ falls inside the K-Nearest-Neighbours of prototypes and feature memory $Q$
 7:     Compute target loss for generated pseudo-labels using Eq. 4
 8:     For unlabeled target samples, compute the contrastive loss between $\tilde{f}$ and $f'$ and the features in the memory bank $Q$ using Eq. 5 and 6.
 9:     Compute the entropy loss for unlabeled target samples to update the prototypes using Eq. 7
10:     Use a gradient reversal layer before the classifier layer to compute the gradients and update the parameters for $\hat{\theta}_F$ and $\hat{\theta}_C$ with Eq. 9
11:     Update the momentum encoder parameters in Eq. 1
12:     Add features and predictions to memory $Q$
13: **until** Training ends
14: **return**  Trained encoder $F$ and classifier $C$

---

## 3.2   Discussion

A key difference between our approach and approaches such as Berthelot et al. (2021) is the generation of pseudo-labels using prototypes based on the embedding instead of the predictions. Recent approaches like AdaMatch Berthelot et al. (2021) suffer from the imbalanced distribution during the generation of pseudo-labels. AdaEmbed selects k nearest neighbors for each prototype to generate a more uniform distribution of pseudo-labels. Additionally, AdaEmbed utilizes the memory bank for the features instead of only passing the gradient of strongly augmented version of the sample.

In addition to the pseudo-labeling strategy, AdaEmbed also incorporates instance contrastive alignment, which has shown promising results in recent domain adaptation literature. Like CLDA Singh (2021), AdaEmbed uses the instance constrastive alignment, but it differs from CLDA by using the memory bank for the features. Moreover, compare to ECACL Li et al. (2021) AdaEmbed uses new categorical alignment loss and instance contrastive alignment. Finally, the main difference between AdaContrast Chen et al. (2022) and AdaEmbed is that the latter uses a new pseudo-labeling strategy based on estimated prototype that are more accurate and more balanced compared to prior works.

# 4   Experiments

## 4.1   Datasets

The performance of our proposed approach, AdaEmbed, is evaluated using several benchmark datasets for domain adaptation. The primary dataset used in our experiments is a subset of the original Domain-Net dataset, referred to as DomainNet-126 Peng et al. (2019). This subset includes 126 classes across 4 domains, namely Real, Sketch, Clipart, and Painting. Our method is evaluated on seven domain shifts, following the experimental settings outlined by Saito et al. (2019). Additionally, we utilize the Office-Home dataset Venkateswara et al. (2017) that includes 4 domains (Real, Clipart, Art, Product) encompassing 65 classes and is widely regarded as a benchmark dataset for unsupervised domain adaptation. Furthermore, we incorporate the VisDA-C dataset Peng et al. (2017), in which we adapt synthetic images to real images across 12 object classes.

## 4.2 Implementation Details

In our experimental framework, the Swin Transformer V2 Liu et al. (2022) serves as the backbone for feature extraction. We utilize the "tiny" variants of the model, closely adhering to the hyperparameters outlined in their study. Prior to the experiments, these Swin Transformer models are pre-trained on the ImageNet-1K dataset Deng et al. (2009) ensuring a solid foundation for subsequent adaptations. For classification tasks, we equip each model with a singular linear head.

Throughout the experimentation process, we organize our data into three distinct mini-batches. The first batch consists of labeled samples from the source domain, the second batch is composed of labeled samples from the target domain, and the third batch encompasses unlabeled samples from the target domain. We incorporate a RandAugment Cubuk et al. (2020) as our data augmentation strategy. Additionally, random erasing is employed to further diversify the training dataset.

For the optimization process, we utilize a Stochastic Gradient Descent (SGD) solver with a base learning rate of 0.05, following a cosine learning rate policy. The experiments are conducted over a maximum of 50 epochs, maintaining a momentum of 0.9 and a weight decay of $10^{-4}$.

For hyperparameters in this method, $\lambda$ is set at 0.2, $\lambda_t$ at 2.0, and $\lambda_c$ at 0.1. The temperature parameter is set to 0.05, with an Exponential Moving Average (EMA) of $m = 0.95$. We use $k = 10$ neighbors and $\tau = 0.9$, to effectively generate pseudo labels. Additionally, a memory queue of size M=1000 is utilized to store features and predictions, facilitating the generation of pseudo-labels and computation of contrastive loss.

All experiments were conducted using the same source code and identical augmentation and hyperparameter settings. This approach ensures consistency across all tests, significantly improving the fairness and reliability of the experiments. Our unified framework, designed for simplicity and comprehensiveness, allows for direct comparisons between methods and a clear understanding of their individual contributions to the overall performance.

## 4.3 Baselines

Our AdaEmbed method is compared with the following domain adaptation strategies:

- **Supervised only**: This approach forms our basic comparison baseline, relying exclusively on labeled data during training.

- **Minimax Entropy (MME)** Saito et al. (2019): Specially designed for SSDA scenarios, MME employs adversarial training to align features and estimate prototypes. It introduces a minimax game where the feature extractor and classifier are trained in tandem to optimize domain-invariant feature representations.

- **Contrastive Learning for Semi-Supervised Domain Adaptation (CLDA)** Singh (2021): CLDA applies contrastive learning to reduce both inter- and intra-domain discrepancies, utilizing inter-domain and instance contrastive alignment.

- **Enhanced Categorical Alignment and Consistency Learning (ECACL)** Li et al. (2021): ECACL leverages strong data augmentation to enhance alignment of category-level features between source and target domains, targeting SSDA scenarios.

- **AdaMatch** Berthelot et al. (2021): AdaMatch represents a unified model that is applicable to Semi-Supervised Learning (SSL), UDA, and SSDA. It accounts for distributional shifts between source and target domains and combines techniques from SSL and UDA to achieve effective domain adaptation.

- **AdaContrast** Chen et al. (2022): Specifically designed for test-time domain adaptation in UDA settings, AdaContrast refines pseudo-labels via soft voting among nearest neighbors in the target feature space.

| Setting | Method | R → C | R → P | P → C | C → S | S → P | R → S | P → R | Avg. |
|---------|--------|-------|-------|-------|-------|-------|-------|-------|------|
| UDA | Source | 62.31 | 66.43 | 60.38 | 57.68 | 56.79 | 51.63 | 75.93 | 61.59 |
| | MME | 67.14 | 70.14 | 62.28 | 57.55 | 61.63 | 59.00 | 75.55 | 63.46 |
| | CLDA | 70.18 | 70.92 | 67.10 | 63.67 | 65.28 | 63.26 | 77.91 | 68.33 |
| | ECACL | 78.30 | 78.54 | 74.04 | 35.02 | 72.68 | 73.03 | 78.78 | 70.05 |
| | AdaMatch | 77.97 | 78.33 | 74.27 | 69.44 | 73.97 | 73.78 | 80.20 | 75.42 |
| | AdaContrast | 74.19 | 76.57 | 72.51 | 68.40 | 67.95 | 70.78 | 79.87 | 72.90 |
| | **AdaEmbed** | 79.14 | 78.22 | 74.37 | 70.01 | 74.87 | 72.53 | 82.69 | **75.98** |
| 1-shot | S+T | 63.47 | 67.53 | 64.08 | 58.04 | 59.48 | 54.72 | 76.91 | 63.46 |
| | MME | 71.53 | 71.44 | 69.63 | 64.47 | 67.33 | 62.59 | 77.78 | 69.25 |
| | CLDA | 71.86 | 71.80 | 70.59 | 64.68 | 67.95 | 64.23 | 79.99 | 70.15 |
| | ECACL | 80.22 | 78.97 | 75.53 | 71.82 | 72.77 | 74.27 | 82.73 | 76.62 |
| | AdaMatch | 79.77 | 78.86 | 75.38 | 72.67 | 75.70 | 72.65 | 82.69 | 76.82 |
| | AdaContrast | 78.18 | 77.65 | 76.94 | 71.95 | 74.26 | 73.30 | 84.25 | 76.65 |
| | **AdaEmbed** | 80.08 | 78.82 | 76.16 | 72.29 | 75.23 | 74.26 | 84.51 | **77.34** |
| 3-shot | S+T | 70.09 | 70.35 | 71.93 | 66.77 | 68.63 | 61.29 | 80.34 | 69.92 |
| | MME | 74.51 | 73.62 | 73.36 | 67.77 | 70.80 | 67.23 | 79.91 | 72.46 |
| | CLDA | 74.43 | 73.08 | 74.09 | 68.01 | 70.79 | 66.41 | 81.57 | 72.63 |
| | ECACL | 82.37 | 79.81 | 79.24 | 73.46 | 77.20 | 75.74 | 84.74 | 78.94 |
| | AdaMatch | 80.68 | 79.11 | 77.30 | 73.76 | 77.07 | 75.11 | 84.95 | 78.28 |
| | AdaContrast | 81.39 | 79.15 | 78.90 | 73.18 | 77.96 | 75.25 | 85.25 | 78.72 |
| | **AdaEmbed** | 81.30 | 79.69 | 78.86 | 74.10 | 77.53 | 75.64 | 85.69 | **78.97** |

Table 1: Performance of AdaEmbed on the DomainNet-126 in UDA and SSDA (1-shot and 3-shot) settings.

### 4.4 DomainNet Experiments

In our evaluation on the DomainNet dataset, AdaEmbed has showcased remarkable performance in both UDA and SSDA settings, as detailed in Table 1. AdaEmbed consistently outperformed all the baselines, demonstrating its robustness and adaptability across varied domain adaptation tasks.

In UDA scenarios, AdaEmbed significantly surpassed the supervised baseline in various domain pairs. For instance, in the Real to Clipart (R to C) adaptation, AdaEmbed achieved an impressive accuracy of 79.14%, a marked improvement over the source-only method's 62.31%. Similarly, in the Painting to Real (P to R) shift, AdaEmbed attained 82.69% accuracy, outperforming all baselines. In SSDA settings, particularly in one-shot and three-shot scenarios, AdaEmbed's effectiveness was further highlighted. Notably, in the challenging Clipart to Sketch (C to S) task, AdaEmbed reached accuracies of 72.29% and 74.10% for one-shot and three-shot settings, respectively, outperforming all the baselines. These results illustrate AdaEmbed's comprehensive approach to domain adaptation.

### 4.5 Office-Home Experiments

The effectiveness of AdaEmbed was rigorously evaluated on the Office-Home dataset, a challenging benchmark for domain adaptation. This dataset, encompassing diverse domains and a wide array of classes, offers a robust platform for testing the versatility of domain adaptation techniques. As Table 2 illustrates, AdaEmbed showcased superior performance in both UDA and SSDA settings.

In UDA, AdaEmbed notably outperformed the Supervised baseline. For instance, in the Real to Clipart (R to C) adaptation, AdaEmbed attained an accuracy of 64.51%, significantly higher than the baseline's 49.22%. Similarly, in the Art to Clipart (A to C) transition, AdaEmbed's accuracy reached 56.34%, exceeding the baseline's 43.15%. In the SSDA context, particularly within one-shot and three-shot frameworks, AdaEmbed's proficiency in leveraging limited labeled data in the target domain was evident. In the one-shot

| Setting | Method | R → C | R → P | R → A | P → R | P → C | P → A | A → P | A → C | A → R | C → R | C → A | C → P | Avg. |
|---------|--------|-------|-------|-------|-------|-------|-------|-------|-------|-------|-------|-------|-------|------|
| UDA | Source | 49.22 | 80.47 | 68.46 | 74.13 | 40.73 | 56.17 | 68.85 | 43.15 | 76.90 | 67.82 | 56.58 | 65.92 | 62.37 |
| | MME | 59.18 | 83.22 | 72.49 | 78.12 | 49.82 | 60.40 | 70.09 | 50.55 | 78.21 | 71.47 | 62.05 | 69.08 | 67.06 |
| | CLDA | 54.12 | 80.74 | 69.49 | 75.50 | 45.65 | 58.80 | 69.82 | 47.73 | 76.79 | 70.00 | 60.16 | 67.18 | 64.67 |
| | ECACL | 63.39 | 85.95 | 76.60 | 81.88 | 56.32 | 69.16 | 76.10 | 53.96 | 80.23 | 73.74 | 64.27 | 74.41 | 71.33 |
| | AdaMatch | 63.44 | 85.18 | 76.03 | 83.19 | 57.72 | 68.01 | 74.73 | 56.34 | 80.16 | 77.02 | 67.19 | 75.92 | 72.08 |
| | AdaContrast | 59.48 | 85.14 | 76.97 | 81.01 | 52.88 | 71.83 | 75.59 | 49.84 | 78.92 | 74.79 | 69.00 | 74.95 | 70.87 |
| | **AdaEmbed** | 64.51 | 86.19 | 75.90 | 82.48 | 55.86 | 68.71 | 77.61 | 56.34 | 82.29 | 77.71 | 69.24 | 77.18 | **72.84** |
| 1-shot | S+T | 54.35 | 83.29 | 72.66 | 78.39 | 47.53 | 61.31 | 77.70 | 51.58 | 78.21 | 73.10 | 63.32 | 74.86 | 68.03 |
| | MME | 61.79 | 85.92 | 75.95 | 80.21 | 56.66 | 65.91 | 79.93 | 56.85 | 80.39 | 75.16 | 68.75 | 78.72 | 72.19 |
| | CLDA | 58.01 | 84.30 | 73.19 | 78.78 | 49.95 | 63.86 | 78.63 | 54.46 | 78.94 | 73.65 | 65.42 | 76.24 | 69.62 |
| | ECACL | 66.21 | 87.70 | 78.37 | 84.36 | 59.75 | 72.53 | 83.36 | 59.34 | 82.80 | 78.14 | 69.70 | 81.58 | 75.32 |
| | AdaMatch | 64.26 | 87.41 | 79.48 | 84.45 | 60.51 | 75.45 | 82.25 | 59.32 | 82.27 | 80.11 | 72.41 | 82.03 | 75.83 |
| | AdaContrast | 63.35 | 86.73 | 79.03 | 83.72 | 61.33 | 75.99 | 82.45 | 59.94 | 82.18 | 78.56 | 72.33 | 81.71 | 75.61 |
| | **AdaEmbed** | 66.60 | 87.77 | 77.92 | 84.31 | 59.66 | 74.42 | 83.36 | 60.23 | 83.99 | 79.70 | 72.33 | 82.97 | **76.11** |
| 3-shot | S+T | 63.58 | 86.04 | 76.56 | 80.07 | 58.26 | 69.04 | 82.64 | 60.37 | 81.51 | 78.10 | 70.07 | 80.02 | 73.85 |
| | MME | 67.83 | 86.89 | 78.78 | 82.57 | 63.55 | 73.07 | 83.85 | 63.67 | 82.36 | 79.84 | 72.78 | 84.08 | 76.61 |
| | CLDA | 63.85 | 86.08 | 75.70 | 80.99 | 58.38 | 69.49 | 82.12 | 61.54 | 80.92 | 78.12 | 71.46 | 80.65 | 74.11 |
| | ECACL | 70.51 | 89.50 | 81.46 | 85.09 | 66.60 | 77.14 | 86.22 | 64.58 | 83.94 | 81.56 | 74.75 | 86.01 | 78.95 |
| | AdaMatch | 69.89 | 88.15 | 80.59 | 85.87 | 65.29 | 77.59 | 85.74 | 64.77 | 84.56 | 82.16 | 77.01 | 85.34 | 78.91 |
| | AdaContrast | 68.32 | 88.11 | 80.43 | 83.74 | 66.55 | 78.00 | 85.25 | 62.93 | 83.46 | 83.12 | 76.11 | 85.77 | 78.48 |
| | **AdaEmbed** | 69.92 | 89.26 | 79.81 | 85.92 | 66.41 | 75.78 | 84.95 | 67.45 | 84.77 | 82.80 | 76.69 | 85.77 | **79.13** |

Table 2: Experiment results on Office-Home dataset across both UDA and SSDA settings.

scenario, AdaEmbed achieved 66.60% accuracy in R to C split and 60.23% in A to C split. Progressing to three-shot adaptation, the accuracies improved to 69.92% and 67.45% respectively. This demonstrates AdaEmbed's effectiveness in utilizing sparse labeled data to enhance adaptability and accuracy in varied domain adaptations.

## 4.6 VisDA-C Experiments

AdaEmbed's capabilities were rigorously evaluated on the VisDA-C dataset, a key benchmark for domain adaptation that concentrates on the transition from synthetic to real-world imagery. This dataset's scope, covering 12 distinct object classes, provided a comprehensive platform for detailed analysis, including class-wise accuracies, average accuracy, and overall performance metrics. The results, detailed in Table 3, demonstrate AdaEmbed's remarkable proficiency in both UDA and SSDA frameworks.

In UDA scenarios, AdaEmbed's performance was particularly notable. It consistently outstripped the achievements of other methods. For example, in the challenging category of trucks, AdaEmbed achieved an accuracy of 58%, which was a significant improvement over the supervised baseline's 12% and higher than AdaContrast's 56%. Similarly, in the motorcycle category, AdaEmbed's accuracy of 95% was comparable to AdaMatch's 95% and ECACL's 97%. These results highlight AdaEmbed's capability to effectively utilize limited labeled data in the target domain.

## 4.7 Ablation Studies

The ablation study shown in Table 4 investigates the contributions of different components in the AdaEmbed method, specifically focusing on their impact on the VisDA-C experiments in UDA setting. The first scenario in our study, "No pseudo-labeling," omits the use of pseudo-labels during training. The results, showing a drop in accuracy to 70.62%, underscore the significance of pseudo-labeling in enhancing the model's performance for unsupervised domain adaptation. When the contrastive loss component is removed, as shown in the second row, there is a noticeable but relatively moderate decrease in accuracy to 85.04%. This result suggests that while the contrastive loss contributes to the overall efficacy of AdaEmbed, its impact may not be as critical as that of the other components. The third row, "No entropy loss," highlights the substantial role of entropy loss in AdaEmbed. Omitting this component leads to a significant reduction in accuracy to 83.43%. This observation indicates that entropy loss is essential in AdaEmbed's methodology, particularly in ensuring the effective alignment of domain-specific features. Finally, the complete AdaEmbed method,

| Setting | Method | plane | bcycl | bus | car | horse | knife | mcycl | person | plant | sktbrd | train | truck | Avg. | Acc. |
|---------|--------|-------|-------|-----|-----|-------|-------|-------|--------|-------|--------|-------|-------|------|------|
| UDA | Source | 90 | 50 | 64 | 76 | 90 | 10 | 94 | 32 | 85 | 29 | 93 | 12 | 60.42 | 65.57 |
|  | MME | 92 | 29 | 33 | 65 | 75 | 4 | 66 | 16 | 98 | 8 | 81 | 2 | 47.42 | 52.37 |
|  | CLDA | 96 | 72 | 85 | 73 | 97 | 19 | 95 | 35 | 94 | 49 | 91 | 14 | 68.33 | 71.08 |
|  | ECACL | 97 | 80 | 91 | 60 | 98 | 18 | 97 | 54 | 96 | 55 | 91 | 14 | 70.92 | 72.46 |
|  | AdaMatch | 98 | 88 | 90 | 75 | 98 | 95 | 95 | 65 | 96 | 84 | 93 | 51 | 85.67 | 83.74 |
|  | AdaContrast | 97 | 90 | 85 | 74 | 97 | 96 | 93 | 84 | 94 | 43 | 92 | 56 | 83.42 | 82.92 |
|  | **AdaEmbed** | 98 | 86 | 88 | 74 | 97 | 94 | 95 | 82 | 96 | 86 | 91 | 58 | **87.08** | **85.17** |
| 1-shot | S+T | 92 | 76 | 93 | 63 | 96 | 31 | 91 | 54 | 95 | 54 | 80 | 6 | 69.25 | 69.88 |
|  | MME | 90 | 69 | 89 | 77 | 93 | 17 | 92 | 78 | 97 | 32 | 69 | 1 | 67.0 | 70.68 |
|  | CLDA | 95 | 79 | 92 | 68 | 97 | 50 | 93 | 56 | 96 | 57 | 85 | 9 | 73.08 | 73.39 |
|  | ECACL | 97 | 82 | 91 | 76 | 98 | 71 | 96 | 69 | 95 | 42 | 90 | 14 | 76.75 | 77.78 |
|  | AdaMatch | 97 | 85 | 90 | 72 | 98 | 94 | 96 | 60 | 97 | 87 | 93 | 50 | 84.92 | 82.76 |
|  | AdaContrast | 98 | 90 | 84 | 69 | 97 | 96 | 92 | 50 | 96 | 1 | 92 | 62 | 77.25 | 78.39 |
|  | **AdaEmbed** | 97 | 89 | 85 | 75 | 97 | 95 | 94 | 84 | 94 | 86 | 92 | 54 | **86.83** | **85.05** |
| 3-shot | S+T | 95 | 80 | 77 | 74 | 94 | 74 | 88 | 40 | 90 | 50 | 87 | 39 | 74.0 | 74.68 |
|  | MME | 95 | 71 | 74 | 84 | 93 | 74 | 95 | 69 | 94 | 41 | 88 | 22 | 75.0 | 76.64 |
|  | CLDA | 96 | 84 | 83 | 73 | 96 | 77 | 92 | 39 | 93 | 60 | 90 | 42 | 77.08 | 76.98 |
|  | ECACL | 97 | 82 | 89 | 80 | 98 | 90 | 96 | 75 | 96 | 82 | 92 | 40 | 84.75 | 83.80 |
|  | AdaMatch | 98 | 86 | 90 | 77 | 98 | 94 | 96 | 59 | 96 | 89 | 94 | 52 | 85.75 | 84.04 |
|  | AdaContrast | 98 | 90 | 86 | 68 | 96 | 95 | 92 | 61 | 94 | 88 | 92 | 60 | 85.0 | 82.34 |
|  | **AdaEmbed** | 98 | 88 | 87 | 82 | 98 | 93 | 96 | 70 | 96 | 89 | 94 | 56 | **87.25** | **85.88** |

Table 3: UDA and SSDA Results on VisDA-C Dataset

| Ablation | $\mathcal{L}_t$ | $\mathcal{L}_c$ | $\mathcal{H}$ | Avg. | Acc. |
|----------|-----------------|-----------------|---------------|------|------|
| No pseudo-labeling |  | ✓ | ✓ | 66.92 | 70.62 |
| No contrastive loss | ✓ |  | ✓ | 86.41 | 85.04 |
| No entropy loss | ✓ | ✓ |  | 86.34 | 83.43 |
| AdaEmbed | ✓ | ✓ | ✓ | **87.08** | **85.17** |

Table 4: Ablation Studies on VisDA-C Dataset

integrating all three losses ($\mathcal{L}_t$, $\mathcal{L}_c$, and $\mathcal{H}$), achieves the best performance, with an average accuracy of 87.08% and overall accuracy of 85.17% on the VisDA-C dataset. This outcome demonstrates the collective strength of these components in AdaEmbed, validating their synergistic impact on enhancing the model's domain adaptation capabilities.

## 4.8 Varying Number of Labels

In the context of semi-supervised domain adaptation, we analyzed the performance impact of a varying number of labeled examples on the OfficeHome dataset Real to Clipart split, as depicted in Figure 3. Our AdaEmbed method consistently outperformed the Supervised Learning and AdaContrast approaches across different quantities of labeled data, notably excelling with fewer labels. The trend suggests a significant initial gain in accuracy with an increase in labeled examples up to 5 per class, beyond which the rate of improvement moderates. While the Supervised method shows a rapid ascent initially, it plateaus, implying a limited benefit from additional data. In contrast, AdaEmbed demonstrates its robustness and efficiency in utilizing sparse labeled data, a critical advantage when such data is a scarce resource, thereby endorsing its adaptability and effectiveness in domain adaptation tasks.

## 4.9 Feature Visualization

In the unsupervised domain adaptation (UDA) setting on the DomainNet dataset's Real to Clipart split, t-SNE visualizations of feature embeddings from four domain adaptation strategies—Supervised Learning (Source Only), AdaMatch, AdaContrast, and AdaEmbed—are analyzed. The Supervised method demonstrates distinct clusters with some overlap, indicating limited domain adaptation. AdaMatch shows tighter

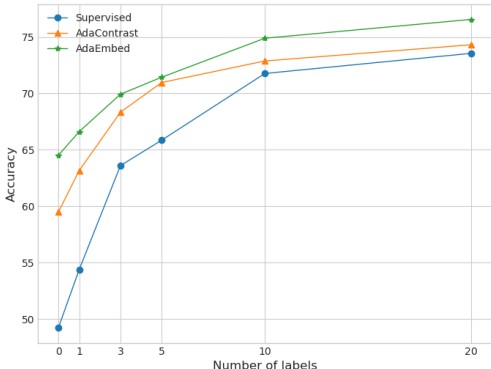

Figure 3: The impact of varying the number of labeled examples on the accuracy of domain adaptation methods in the OfficeHome dataset's Real to Clipart split. The plot demonstrates the performance of three strategies: Supervised Learning, AdaContrast, and AdaEmbed. Each method's accuracy improves with more labeled data, with AdaEmbed showing the highest efficiency and performance across all data regimes, especially in settings with fewer labeled examples.

clustering, yet with notable inter-class mixing. AdaContrast's approach yields clearer class demarcation, signifying better domain invariance. Our proposed AdaEmbed method outshines the others, presenting the most compact and well-separated clusters, indicative of superior domain adaptation by effectively aligning domain-invariant features and minimizing within-class variance. These embeddings, as visualized, validate AdaEmbed's robustness in learning transferable features across domains.

## 5  Conclusion

In this study, we introduced AdaEmbed, a novel and efficient approach for semi-supervised domain adaptation (SSDA) that successfully navigates the challenges posed by domain shift in computer vision tasks. AdaEmbed is designed to leverage the synergy between a shared embedding space and prototype generation, facilitating the creation of accurate and balanced pseudo-labels for unlabeled target domains. This approach effectively harnesses cross-entropy loss and contrastive loss as dual sources of supervision, addressing the imbalance in pseudo-label distribution—a common challenge in SSDA.

Our extensive evaluation of AdaEmbed across prominent domain adaptation benchmarks, such as DomainNet-126, Office-Home, and VisDA-C, has demonstrated its remarkable efficacy in both SSDA and unsupervised domain adaptation (UDA) settings. The results consistently indicate AdaEmbed's superiority over existing state-of-the-art methods, establishing new benchmarks across a range of evaluation metrics. Notably, AdaEmbed's innovative pseudo-labeling strategy and its robustness against distribution shifts in the embedding space set it apart from existing methods. AdaEmbed emerges as a robust and practical solution for domain adaptation in real-world scenarios, where labeled data is often scarce or unevenly distributed. By addressing both theoretical and practical aspects of SSDA, AdaEmbed paves the way for future research in semi-supervised domain adaptation.

### Broader Impact Statement

This research on AdaEmbed for semi-supervised domain adaptation primarily offers positive advancements in computer vision, particularly in efficiently handling domain shifts with limited data. Its model-agnostic design enhances its applicability across various sectors and promotes resource efficiency, especially in settings with limited computational resources. However, it is crucial to be mindful of potential negative impacts, such as the misuse of the technology in privacy-sensitive applications like surveillance, the risk of reduced human oversight in critical decision-making processes, and the perpetuation of inherent data biases. As such, while AdaEmbed presents significant benefits, its ethical use, continuous evaluation for fairness, and maintenance of human oversight are essential to ensure its responsible and beneficial application in society.

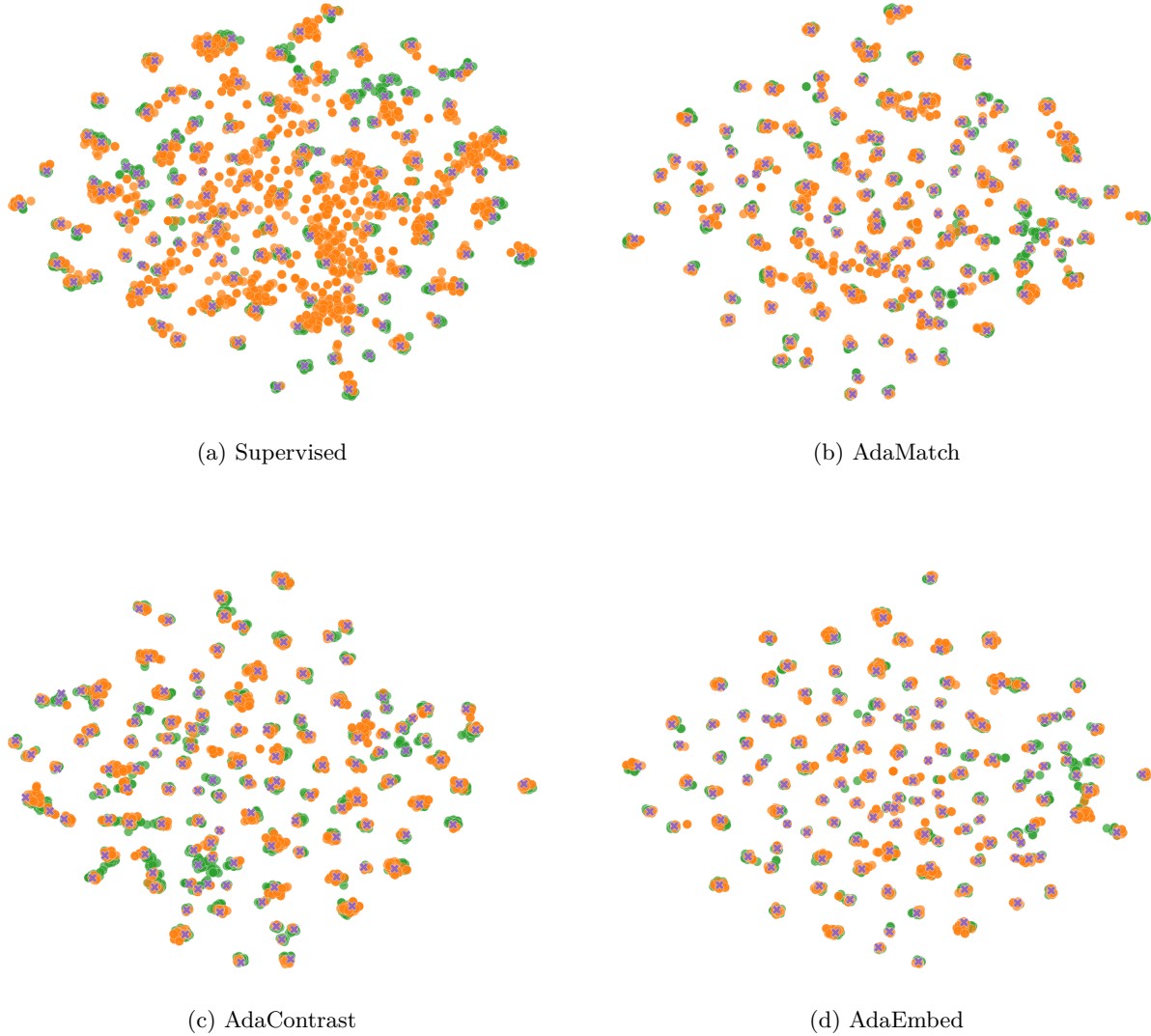

(a) Supervised

(b) AdaMatch

(c) AdaContrast

(d) AdaEmbed

Figure 4: Comparative t-SNE visualizations of feature embeddings from different domain adaptation methods on the DomainNet Real to Clipart split under unsupervised domain adaptation setting. Green and orange circles represent the source and target domains, respectively, while purple crosses mark the prototypes. The Supervised Learning method exhibits basic clustering with overlaps, reflecting a foundational level of domain adaptation. AdaMatch and AdaContrast display progressively tighter and more distinct clusters, signaling advancements in domain invariance. However, AdaEmbed stands out distinctly, showcasing highly compact and distinctly separated clusters. These embeddings visually encapsulate the superior domain adaptation proficiency of AdaEmbed.

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
