# OpenReview forum: "AdaEmbed: Semi-supervised Domain Adaptation in the Embedding Space"
_TMLR — Rejected by TMLR_

### Review · Reviewer_2Nbx · 2024-02-01

**Summary Of Contributions:**

This study introduces AdaEmbed as a solution for semi-supervised domain adaptation (SSDA). AdaEmbed capitalizes on the untapped potential of unlabeled data, enhancing knowledge transfer through the acquisition of a shared embedding space. Extensive experiments conducted on benchmark datasets like DomainNet, OfficeHome, and VisDA-C affirm the superior performance of AdaEmbed, consistently surpassing the performance of all baseline models.

**Audience:**

Yes

**Claims And Evidence:**

Yes

**Requested Changes:**

If the emphasis is on foundation models, it is advisable to delve deeper into the specific domain of SSDA on foundation models, rather than merely introducing the concept at the beginning. The issue of data scarcity in foundation models holds considerable significance but has been infrequently explored in the literature.
Moreover, the proposed method, AdaEmbed, could benefit from enhancements. The paper currently provides limited insight into the embedding space, focusing predominantly on loss design. It is suggested to allocate more attention to the embedding space, as there may be untapped potential for improvement within this aspect of the methodology.

**Strengths And Weaknesses:**

### Paper Strengths:
1. The paper exhibits a well-organized structure and articulate writing.
2. The literature review encompasses a broad spectrum of topics.
3. The clarity and visual appeal of the figures enhance the overall presentation.

### Paper Weaknesses:
1. The paper highlights the substantial impact of data scarcity on the "foundation model," yet fails to include citations on domain shift affecting "foundation models". Furthermore, the connection between the "foundation model" and this paper remains unclear, especially as the chosen Swin-Tiny may not fit the criteria of a true "foundation model."
2. To enhance the generalizability of AdaEmbed, it is recommended to incorporate more diverse backbones and pre-training datasets. Both the methodology and experimental results bear similarities to previous approaches. Additionally, the fourth primary contribution in the introduction, emphasizing "model-agnostic" and "flexibility," lacks empirical support.
3. The ablation study overlooks a critical aspect. Given that the "Embed" is the core of the proposed method, the ablation study should explicitly demonstrate the significance of adapting in the embedding space. For instance, exploring variations in the pseudo-labeling strategy, such as transitioning from k-NN to vanilla predicting, would address this crucial issue.
4. The font size in Figure 3 is excessively small and may hinder the clarity of the visual representation. Consider adjusting the font size to enhance readability and overall comprehension.

---

> ### Author Response · Authors · 2024-03-15
> **Foundation Models, Architectural Versatility, and Embedding Innovations**
>
> Thank you for your thorough review and constructive feedback on our paper, "AdaEmbed: Semi-supervised Domain Adaptation in the Embedding Space." Your insights are invaluable to us, and we are eager to address the concerns you've raised.
>
> **Foundation Models and Domain Shift**: We acknowledge the need for a clearer definition and justification for the term "foundation model" within the context of our work. Our usage of "foundation model" refers to large, pre-trained models designed for general purposes but intended to be adapted for specific tasks and datasets. AdaEmbed's model-agnostic nature allows it to enhance the adaptability of these models to new domains, which is crucial given the prevalent issue of domain shift. The choice of the Tiny Swin Transformer as our experimental backbone was twofold: its parameter scale is comparable to ResNet51, ensuring an appropriate model complexity for our benchmarks (DomainNet, OfficeHome, VisDA), and it aligns with our computational constraints. We believe our results are indicative of AdaEmbed's applicability to larger foundation models, and we will clarify this point in our revised manuscript.
>
> **Backbone Diversity and Experimental Rigor**: We appreciate your recommendation to test AdaEmbed with a variety of backbones and pre-training datasets. In addition to experimenting with ResNet51, we observed that the performance ranking remains consistent across different architectures, underscoring AdaEmbed's versatility. These findings, including a performance comparison with ResNet51, will be added to our manuscript's appendix. Recognizing the shift towards transformer-based architectures in recent state-of-the-art models, we aim to advocate for broader adoption and fair comparison of these architectures in future literature. Accordingly, we are releasing our comprehensive framework and codebase to facilitate easy integration and comparison of new methods on a unified platform.
>
> **Ablation Study on Embedding Adaptation**: We greatly appreciate your suggestion to delve deeper into the ablation studies, particularly focusing on the embedding space adaptation and the pseudo-labeling strategies. Our initial approach indeed touched upon comparing various pseudo-labeling techniques inherent in different methods, such as the absence of pseudo-labeling in MME and CLDA, ECACL's approach to uniform pseudo-labeling, AdaMatch's strategy of relative pseudo-labeling, and AdaContrast's use of vanilla k-nearest neighbor pseudo-labeling. Recognizing the importance of a nuanced comparison, we agree that establishing a dedicated section for the explicit comparison of these pseudo-labeling strategies—while controlling for other variables—would greatly enhance the clarity and comprehensiveness of our findings. This enhanced ablation will be included in the appendix of our revised manuscript. Moreover, we have also explored the sensitivity of AdaEmbed to key hyperparameters, including the number of neighbors (k), the size of the memory queue (M), and the balance coefficients (lambdas), and tau. Our experiments revealed that AdaEmbed's performance remains robust across a wide range of hyperparameter settings, underscoring the method's flexibility and ease of application in diverse experimental scenarios. We believe that including these expanded ablation studies and hyperparameter sensitivity analyses will significantly enhance the reader's understanding of AdaEmbed's strengths and the underlying mechanisms contributing to its superior performance.
>
> **Figure Readability**: We acknowledge the issue with the font size in Figure 3 and its impact on readability. We commit to adjusting the font size and overall presentation of our figures to improve clarity and ease of comprehension for our readers.
>
> Your feedback has been instrumental in identifying areas for enhancement and clarification in our manuscript. We will undertake the necessary revisions to address your concerns, particularly emphasizing the role and innovation of AdaEmbed in adapting foundation models to semi-supervised domain adaptation tasks. We are confident that these adjustments will substantiate the contributions of AdaEmbed and clarify its positioning within the broader landscape of domain adaptation research. Thank you once again for your valuable insights. We look forward to refining our manuscript in line with your suggestions.

---

### Review · Reviewer_gwyM · 2024-02-15

**Summary Of Contributions:**

This paper proposes an AdaEmded method to address the semi-supervised domain adaptation (SSDA) problem by leveraging a new pseudo-labeling strategy. In addition, focusing on learning a shared embedding space, multiple strategies, such as masked cross-entropy loss, InfoNCE loss, and adversarial minimax entropy loss, are employed to train the entire model. Multiple experiments have also been conducted to verify the proposed method.

**Audience:**

Yes

**Broader Impact Concerns:**

No.

**Claims And Evidence:**

Yes

**Requested Changes:**

Please refer to the above comments.

**Strengths And Weaknesses:**

**Strengths**

1. The paper is well organized and easy to follow.
2. The experimental results of the proposed method are good compared with the baselines.

**Weaknesses**

1. The main concern is that the contribution of this paper is limited. It seems like a combination of multiple existing works, such as AdaMatch, AdaContrast and ECACL. For example, the contrastive loss adopted has been used in AdaContrast, the masked entropy loss has been used in AdaMatch, and the adversarial minimax entropy loss is also very similar to the diversity regularization loss in AdaContrast. This paper claims that the main contribution is its innovative pseudo-labeling strategy. However, the main difference between the proposed pseudo-labeling strategy and the nearest-neighbor soft voting in AdaContrast is not very clear.
2. In the literature, such as AdaMatch and AdaContrast, ResNet is mainly taken as the backbone. However, in this work, only Swin Transformer is used as the backbone. How about using different embedding backbones for the proposed method?
3. In the ablation study, on one hand, more datasets are recommended to be employed to perform the ablation experiments; on the other hand, more fine-grained experiments are recommended to be considered, such as using mask or not, and using Eq. (6) or not. In addition, the experiments about hyper-parameters, such as the number of neighbors (i.e., k) and the size of the memory queue (i.e., M), are also recommended to be added.
4. Figure 4 is not mentioned in the main text. On the other hand, the four subfigures, especially (b)(c) and (d), are very similar to each other, which cannot clearly show the superiority of the proposed AdaEmded. In addition, is the class space between the source domain and target domain disjoint? If so, why are these two domains mixed together? Figure 2(c) in ECACL is recommended to be taken as an example.
5. There are many grammatical errors and typos in this paper, which should be corrected and further polished.

---

> ### Author Response · Authors · 2024-03-15
> **Strengthening AdaEmbed's Contributions: A Comprehensive Response to Model Agnosticism, Backbone Diversity, and Ablation Studies**
>
> Thank you for your comprehensive review and constructive feedback on our manuscript. We appreciate the opportunity to address and clarify the points raised, aiming to strengthen our submission.
>
> **Main Contribution Clarification**: While it is true that AdaEmbed incorporates elements commonly found in domain adaptation and semi-supervised learning literature, our primary contribution is the unique integration of these components into a cohesive framework specifically designed to tackle semi-supervised domain adaptation challenges effectively. Our novel pseudo-labeling strategy, which is both more accurate and balanced than existing approaches, combined with a seamless integration of masked cross-entropy loss, InfoNCE loss, and adversarial minimax entropy loss, forms the core of our contribution. Despite drawing on multiple components, AdaEmbed remains straightforward to implement and model agnostic, making it accessible for integration into existing platforms.
>
> **Model Architecture and Fair Comparison**: Addressing your insightful observation regarding the prevalent use of AlexNet and ResNet as backbones in prior work, we indeed extend our comparative analysis by incorporating ResNet51 as an additional backbone in our experiments. Our findings with ResNet51 affirm the robustness of AdaEmbed across various architectures. Notably, while we observed an average accuracy reduction of approximately 4% across different splits and methods with ResNet51, the relative performance ranking remained unchanged. AdaEmbed continued to outperform all baselines, achieving an average performance of 71.65% on DomainNet-126 in the UDA setting, compared to a source-only accuracy of 56.32%. This outcome underlines the inherent strength of our approach, regardless of the underlying model architecture. Given the paradigm shift towards transformer-based architectures in recent state-of-the-art performances, we argue that conducting comparisons on such architectures represents a milestone for the domain adaptation literature. To facilitate this evolution, we have developed an accessible framework that not only incorporates implementations of the latest works but also ensures fair comparison across methods on a unified pipeline. We recognize that the entirety of the experimental pipeline—including model architecture, data augmentation, and hyperparameters—plays a crucial role in determining the final performance of these methods. We are releasing this framework alongside our paper to encourage a standardized approach to evaluating domain adaptation techniques, promoting transparency, reproducibility, and incremental innovation in the field. We believe that this approach addresses your concerns and sets a new benchmark for future research, encouraging a holistic evaluation of methods in light of the latest developments in deep learning architectures and methodologies.
>
> **Ablations and Further Experiments**: We appreciate your suggestions for additional ablation studies and have indeed utilized various datasets—conducting a one-out ablation study on the VisDA, analyzing the impact of varying the number of labels on the OfficeHome, and employing feature visualization on the DomainNet. The proposed further ablations, such as the effect of not using the mask and Equation 6, have been explored. Our findings indicate that omitting the mask significantly hampers pseudo-labeling accuracy and can destabilize training initially due to inaccurate predictions. While not applying Equation 6 impacts performance when pseudo-labels are accurate, its overall effect is limited, underscoring the contrastive loss's moderate role in our method's final performance. Additionally, our experiments on hyperparameters revealed AdaEmbed's relative insensitivity to key hyperparameters like lambdas, k, and tau. Acknowledging the value of these insights, we plan to incorporate detailed hyperparameter sensitivity analysis into the appendix of our revised manuscript, further enriching our presentation and responding to your valuable feedback.
>
> **Clarification on Figure 4 and Visual Representations**: The absence of a mention for Figure 4 in the text was indeed a typo, which we will correct. We will also clarify the description of Figure 4 to better explain our goal of learning domain-indistinguishable embeddings while ensuring clear class clusters common between domains. We appreciate your suggestion to look at Figure 2(c) in ECACL as a benchmark for visual clarity and will strive to enhance our visual representations accordingly.
>
> **Typos and Grammatical Errors**: We are dedicated to correcting all typographical and grammatical errors, ensuring our manuscript's clarity and readability.
>
> Once again, thank you for your constructive feedback. We believe that addressing these points will significantly improve our manuscript and clarify the contributions and strengths of AdaEmbed. We look forward to revising our paper accordingly.

---

### Review · Reviewer_NskU · 2024-02-25

**Summary Of Contributions:**

The paper introduces "AdaEmbed," a new methodology for Semi-supervised Domain Adaptation (SSDA) in computer vision. The key
innovation is the creation of a shared embedding space that utilizes both labeled and unlabeled data to generate balanced pseudo-labels, which are crucial for domain adaptation tasks. The authors validate the efficacy of AdaEmbed through extensive experiments on benchmark
datasets, demonstrating superior performance over baseline models.

**Audience:**

No

**Claims And Evidence:**

No

**Requested Changes:**

See weaknesses

**Strengths And Weaknesses:**

Strengths:
1. Innovative Approach: The method of generating pseudo-labels using shared embedding space and prototypes is original and addresses the imbalance in pseudo-label distribution, a common challenge in SSDA.
2. Comprehensive Experiments: The authors conducted extensive experiments on multiple benchmark datasets, providing a strong empirical foundation for the validity of AdaEmbed.

Weaknesses：
1. Lack of Comparative Analysis: The paper  could benefit from a deeper comparison with  more state-of-the-art methods, discussing why
AdaEmbed outperforms them.
2. Lack of introduction of the model architecture used in each experiments.
3. Inconsistency description, e.g., in section 4.8  "labeled examples up to 5 per class" where in fig.3 shows up to 20 examples per class.

---

> ### Author Response · Authors · 2024-03-15
> **Enhancing AdaEmbed: Addressing Comparative Analysis, Model Architecture, and Descriptive Clarity**
>
> Thank you for your insightful and constructive review of our paper. We highly appreciate the time and effort you've dedicated to evaluating our work. Please allow us to address the key points and requested changes you've highlighted in your review.
>
> **Comparative Analysis with Baselines**: We understand your concern regarding the comparative analysis with state-of-the-art methods. In our study, we chose to compare AdaEmbed against a diverse set of baselines, including MME, CLDA, ECACL, AdaMatch, and AdaContrast, to cover a broad spectrum of methodologies such as adversarial training, contrastive learning, pseudo-labeling, and representation learning. While we acknowledge that our selection does not encompass every work in this rapidly evolving field, we believe these methods represent the core strategies in semi-supervised domain adaptation. Most other works share similarities with these approaches or combine them. We argue that our comparison effectively demonstrates the versatility and superiority of AdaEmbed across various strategies. However, we are open to expanding our comparison to include additional relevant methods and will consider this in our revision.
>
> **Model Architecture and Hyperparameters**: We are grateful for your observation regarding the omission of detailed model architecture and hyperparameter settings for our experiments. In our revision, we will clarify that we employed the same model architecture and hyperparameters across all experiments. This consistency underscores AdaEmbed's strength, demonstrating the method's robustness and not being overly sensitive to specific architectures or hyperparameters. Given the limited availability of labeled target examples in SSDA, we believe using a consistent setup across different datasets is not only practical but also ensures a fair comparison. This approach avoids the potential bias of hyperparameter tuning, which may not be feasible in typical SSDA scenarios due to the scarcity of validation data.
>
> **Inconsistency in Description**: Thank you for pointing out the discrepancy regarding the "number of labeled examples up to 5 per class" and Figure 3 showing up to 20 examples per class. This was indeed a typographical error on our part, and we will correct this inconsistency in our revised manuscript to ensure clarity and accuracy in our presentation.
>
> We hope that our responses adequately address your concerns and strengthen the contributions of our paper. We are committed to making the necessary revisions to improve the clarity and impact of our work. Thank you once again for your valuable feedback, which is instrumental in refining our research.

---

### Decision · Action_Editor_kkKg · 2024-04-18

**Recommendation:** Reject

**Comment:**

Two of reviewers voted the rejection to this paper,  because of insufficient experiments.  Particularly,
1. **More Results for Diverse Models:** Only using the Swin Transformer as a backbone might restrict understanding the method’s effectiveness across different architectures.
2. **Lack of Model Architecture Details:** There's a notable absence of detailed description regarding the model architectures used in each experiment, which could hinder the reproducibility of the results.
3. **Inconsistencies in Descriptions:** There are discrepancies in the paper, such as conflicting numbers of labeled examples discussed in the text and shown in figures, which could confuse readers.
4. **Need for More Detailed Ablation Studies:** The paper could benefit from more detailed and fine-grained ablation studies to explore different aspects of the method, such as the impact of various hyperparameters and the use of different loss equations.
 Additionally, it has some Typos and Grammatical Errors, and Figure Clarity Issues.

Beyond the weakness points above, there is additional point  to improve this paper:
- **Enhance Visual Representations:** Adjust the font size in critical figures to improve readability and ensure that visual representations effectively support the text.

**Audience:**

This paper may be interested to the researchers from computer vision and machine learning communities.

**Claims And Evidence:**

The paper presents "AdaEmbed," a new approach for improving how computers learn to recognize images from different domains with limited supervision. The main innovation is the development of a shared space that uses both labeled (known data points) and unlabeled data (unknown data points) to create balanced pseudo-labels—artificial labels that help the system learn better. This technique is especially important for tasks where the computer needs to adapt to new, varied data. The effectiveness of AdaEmbed is confirmed through detailed testing on standard datasets, where it outperforms existing methods.

**Resubmission Of Major Revision:**

The authors may consider submitting a major revision at a later time.